# Latent Chain-of-Thought Improves Structured-Data Transformers

## Abstract

Chain-of-thought and more broadly test-time compute are known to augment the expressive capabilities of language models and have led to major innovations in reasoning. Motivated by this success, this paper explores latent chain-of-thought as well as the impact of depth and looping for time-series and tabular data. We propose a recurrent scheme in which a structured-data transformer, after an initial forward pass, compresses its query-position hidden states into feedback tokens that are appended to the input and processed again, allowing multiple rounds of latent computation before prediction. We compare CoT models against a same-depth no-CoT baseline, a deeper baseline matched to the CoT model in effective depth, and a looped transformer with weight-tied recurrence but no additional chain-of-thought tokens. Across 36 datasets in time-series forecasting and tabular prediction, latent chain-of-thought improves over the baseline on 8/9 time-series datasets (+10.99% average gain) and 22/27 tabular datasets (+5.31% average gain). Across both settings, the CoT models perform the best on average. These results demonstrate that chain-of-thought is a useful axis for scaling test-time compute for structured data.

## 1. Introduction

One of the main vehicles for recent progress in large language models has been scaling test-time compute. Reasoning models such as OpenAI's o1 (OpenAI, 2026), DeepSeek-R1 (Guo et al., 2025), and their successors demonstrate that allocating more computation per query can substitute for, and sometimes outperform, scaling parameters (Snell et al., 2024). One of the main mechanisms for scaling test-time compute is chain-of-thought reasoning: inducing the model

to generate intermediate reasoning tokens in natural language to think through the problem (Wei et al., 2023).

It is not clear whether this reasoning has to happen in natural language. Most tokens in a chain of thought serve textual coherence rather than computation, and decoding to a discrete token at each step discards most of the information in the model's hidden state. (Hao et al., 2025) address this directly with Coconut, which feeds the model's last hidden state back as the next input embedding rather than decoding it to a single word, replacing discrete token-based chain-of-thought with a chain of latent thoughts. (Geiping et al., 2025) achieve a similar goal through recurrent depth: rather than extending the sequence, the model iteratively applies a shared transformer block to refine its internal latent state before producing an output. This enables additional test-time computation without requiring chain-of-thought supervision or longer context. (Gozeten et al., 2025) further show that latent chain-of-thought enables parallel exploration of multiple reasoning paths, demonstrating additional flexibility of this paradigm for test-time scaling. Both of these approaches build on earlier work on weight-tied depth recurrence in the Universal Transformer (Dehghani et al., 2019) and looped transformers (Giannou et al., 2023), as well as papers showing that even content-free tokens can be used to allocate additional forward-pass computation (Goyal et al., 2024).

All of this work has focused on language. Yet the same learning setup (a transformer doing in-context learning) is now used by state-of-the-art models for structured data. In tabular prediction, TabPFN (Hollmann et al., 2023) introduced in-context learning (ICL) over labeled rows, and its successors TabPFNv2 (Grinsztajn et al., 2026) and TabICL (Qu et al., 2025) use alternating row and column attention to scale this paradigm to medium and large tables, challenging the long-standing dominance of gradient-boosted trees (Chen & Guestrin, 2016). In time-series forecasting, transformer-based foundation models such as Chronos (Ansari et al., 2024), TimesFM (Das et al., 2024), and Moirai (Woo et al., 2024), as well as some domain-specific foundation models (Zhu et al., 2025; Dudley et al., 2025), have become the standard. In both tabular prediction and time-series forecasting the model is a transformer that attends over a structured set of observations and produces predictions in a single forward pass—the same setting where adding latent

[1]Anonymous Institution, Anonymous City, Anonymous Region, Anonymous Country. Correspondence to: Anonymous Author <anon.email@domain.com>.

Preliminary work. Under review by the International Conference on Machine Learning (ICML). Do not distribute.

*Figure 1.* **Latent chain-of-thought for structured data.** The transformer $f_\theta$ runs on a sequence of context tokens, query tokens, and (after the first pass) appended feedback tokens. Query-position hidden states $H_q^{(r)}$ are compressed by an MLP $\phi_\theta$ into feedback tokens $Z^{(r)}$, which are appended to the sequence for the next pass. After $R$ recurrences, the prediction head $g_\theta$ maps from the hidden states to the prediction $\hat{y}$.

computation has paid off in language modeling.

In this paper, we ask whether latent chain of thought can transfer to structured data. Our approach gives a structured-data transformer multiple rounds of latent computation before it makes a prediction. After an initial forward pass, we take the hidden states at the prediction positions, compress them through a small MLP into feedback tokens, append those tokens to the model's input representation, and run the transformer again. To enable controlled comparisons, we train each model from scratch on a single dataset rather than pretraining a separate foundation model for each recurrence variant. We compare against (i) a no-recurrence baseline at the same depth, (ii) a deeper baseline matched in total compute, isolating the contribution of latent recurrence from the confounds of pretraining scale and parameter count, and (iii) looped-transformer baselines that share weights across passes but lack the explicit feedback-token mechanism (i.e., recurrent passes with no CoT tokens) (Giannou et al., 2023).

We adapt latent chain of thought to tabular prediction and time-series forecasting, evaluate it on nine time-series datasets from LTSF-9 (Zeng et al., 2022) and a 27-dataset subset of OpenML that was used in TabPFN evaluations (Bischl et al., 2021), and characterize when our approach improves prediction relative to both same-depth and matched-compute baselines.

## 2. Methods

We study whether structured-data transformers can benefit from latent chain-of-thought. Let $x$ denote the structured input, and let $q$ denote the prediction positions: query rows in tabular prediction or future-patch query tokens in time-series forecasting. A standard transformer computes hidden states $H^{(0)} = f_\theta(x)$, and predicts from the hidden states at the query positions using a prediction head, $\hat{y}^{(0)} = g_\theta(H_q^{(0)})$. Our method adds recurrent passes through the same transformer. At recurrence step $r$, we extract the query-position hidden states $H_q^{(r)}$, map

them through a two-layer MLP $Z^{(r)} = \phi_\theta(H_q^{(r)})$, and append the resulting tokens to the embedded input sequence: $E^{(r+1)} = [E^{(0)}; Z^{(0)}; \ldots; Z^{(r)}]$, where $E^{(0)}$ denotes the embedding of $x$. The transformer is then run again on the sequence $H^{(r+1)} = f_\theta\big(x, Z^{(0)}, \ldots, Z^{(r)}\big)$. After $R$ recurrences, the final prediction is decoded from the original query positions, $\hat{y} = g_\theta(H_q^{(R)})$.

For tabular tasks, we use the three-stage architecture of TabICL (Qu et al., 2025). Each scalar cell is embedded as a token, with learned embeddings added to distinguish columns. The input consists of context rows $(X_c, y_c)$ and query rows $X_q$. Context labels are in a label column, while query rows receive a learned no-label token. A column-wise transformer $\text{TF}_{\text{col}}$ first attends across samples within each feature, a row-wise transformer $\text{TF}_{\text{row}}$ then attends across features within each row to produce a single embedding per row, and an in-context learning transformer $\text{TF}_{\text{icl}}$ attends across rows to predict query labels from labeled context rows. Recurrence is applied at the $\text{TF}_{\text{icl}}$ stage: the query positions $q$ are the query-row positions in $\text{TF}_{\text{icl}}$, and feedback tokens are appended to the row-embedding sequence before $\text{TF}_{\text{icl}}$ is rerun. The first two stages are run once per query and their outputs are reused across recurrences. Predictions are trained with cross-entropy for classification and root mean squared error for regression.

For time-series tasks, we use a patch-based forecasting transformer (Nie et al., 2023). A (potentially multivariate) context window $X_{1:T} \in \mathbb{R}^{T \times C}$ is divided into patches and embedded into a sequence of context tokens. We append learned query tokens corresponding to future patches and run a standard transformer over the combined sequence. The hidden states at the future-query positions are decoded into quantile forecasts for the prediction horizon. During chain-of-thought, the future-query hidden states are appended to the input before rerunning the same transformer stack. The final forecast is decoded after $R$ recurrences. We train forecasts using the quantile loss over quantile levels $\{0.1, 0.3, 0.5, 0.7, 0.9\}$.

*Table 1.* Aggregate benchmark performance across structured-data tasks. "Best CoT" uses the CoT depth selected using validation performance from $\{1, 2, 4\}$ independently for each dataset, "Best Looped" uses the same validation approach. Average rank is computed per dataset across baseline, deeper, best CoT, and best looped. Average gain is reported relative to the same-depth baseline, with positive values indicating improvement.

| Domain | Method | Avg. Rank ↓ | Wins vs. Baseline | Avg. Gain vs. Baseline ↑ |
|---|---|---|---|---|
| Time series | Baseline | 2.9 | – | – |
| | Deeper | 3.6 | 3 / 9 | -8.28% |
| | Best CoT | 1.6 | 6 / 7 | 10.99% |
| | Best Looped | 1.9 | 6 / 7 | 3.01% |
| Tabular | Baseline | 3.0 | – | – |
| | Deeper | 2.9 | 15 / 27 | 0.53% |
| | Best CoT | 1.6 | 22 / 27 | 5.31% |
| | Best Looped | 2.5 | 17 / 27 | 1.44% |

We train each model from scratch on each dataset with a fixed chain-of-thought length $R_{\text{train}} \in \{0, 1, 2, 4\}$ and evaluate at multiple recurrence depths $R \in \{0, 1, 2, 4, 8\}$. For each task, we compare against three families of baselines, each designed to isolate a specific aspect of our experiment. The first baseline is a same-depth $L$-layer transformer without CoT ($R = 0$). This controls for parameter count and isolates the effect of additional test-time computation. Next, we train a deeper baseline, a $2L$-layer transformer trained without recurrence, matching a CoT model at $R = 1$ in effective forward-pass depth (and roughly in FLOPs) while doubling the parameter count. Comparing our CoT models to this baseline distinguishes gains from CoT from gains attributable simply to depth or capacity. Finally, we compare to looped baselines that share weights across passes but do not append additional tokens to the input. For these, a stack of $K$ transformer blocks is applied $M$ times for an effective depth of $K \cdot M$. We consider four configurations covering two regimes: weight-tied single-block looping in the style of (Giannou et al., 2023) ($K = 1$, $M \in \{2, 4\}$) and Universal-Transformer-style looping over a multi-block stack (Dehghani et al., 2019) ($K = 4$, $M \in \{2, 4\}$). Comparing CoT against the looped baselines isolates the contribution of the additional CoT tokens specifically, since both methods reuse the same weights across passes. All baselines are trained from scratch with the same optimizer, learning-rate schedule, and epoch budget as the CoT models.

## 3. Results

Table 1 reports aggregate performance across both domains. Selecting the chain-of-thought depth per dataset based on validation performance, latent chain-of-thought improves over the same-depth baseline on 8 out of 9 time-series datasets ($+10.99\%$ average gain in quantile loss) and 22 out of 27 tabular datasets ($+5.31\%$ average gain). The same chain-of-thought models also outperform the depth-matched deeper baseline on average by 21.01% on time series and

4.75% on tabular, indicating that the gains are not simply a consequence of additional compute or parameters: stacking more layers in a non-recurrent transformer yields worse average rank than the original baseline on time series (3.6 vs. 2.9) and only marginal improvement on tabular (2.9 vs. 3.0). This is likely due to overfitting: some of the benchmark datasets are relatively small. Finally, the best CoT model also outperforms the best looped model by 7.74% on time series forecasting and 3.82% on tabular prediction, indicating that both recurrence and explicit chain-of-thought tokens are necessary for maximum benefit. Per-dataset results are in the appendix.

**Scaling with chain-of-thought depth.** Figure 2 shows how performance varies with fixed chain-of-thought depth (each model trained and evaluated at the same $R$), in contrast to the per-dataset best-depth selection used in Table 1. On time series, all tested depths $R \in \{1, 2, 4, 8\}$ improve over the baseline, with average gains between roughly 5.7% and 6.5% in quantile loss and no clear trend in performance versus CoT depth. The depth-8 setting is obtained by training at $R_{\text{train}} = 4$ and increasing the number of recurrences only at evaluation, suggesting that the learned recurrent update generalizes beyond its training horizon. On tabular classification tasks (metric = classification accuracy, in percentage points), the relationship is monotone above $R = 1$: a single recurrence slightly hurts performance ($-1.8$ pp), but additional recurrences recover and surpass the baseline, reaching $+3.1$ pp at $R = 2$ and $+6.7$ pp at $R = 4$. The shape of both curves—monotone improvement with depth on tabular, and a tendency toward diminishing or non-monotone returns on time series—is broadly consistent with observations in language modeling that more chain-of-thought often helps but can plateau or degrade past a point (Yang et al., 2025).

**Latent recurrence vs. added depth and looping.** Comparing chain-of-thought to the deeper and looped baselines isolates the contribution of CoT tokens from raw compute

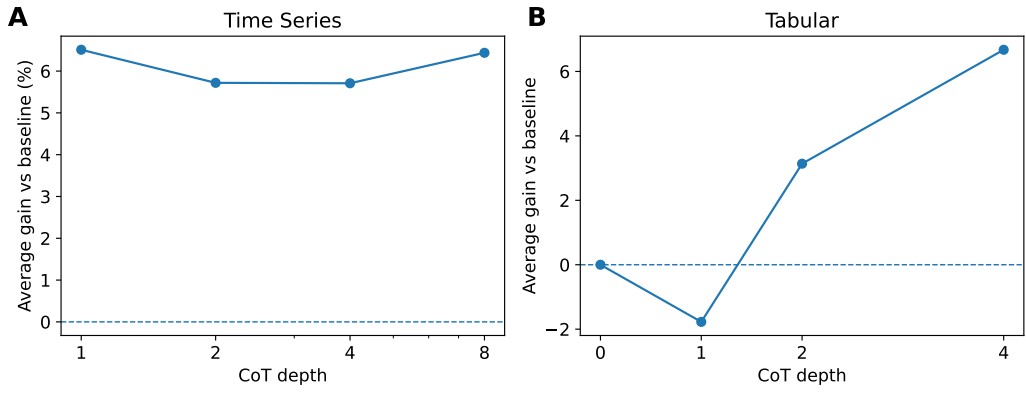

*Figure 2.* **Performance gains from latent chain-of-thought as a function of recurrence depth.** Each point is the mean across datasets of the per-dataset gain over the same-depth no-recurrence baseline at a fixed training/evaluation depth $R$ (in contrast to Table 1, which selects $R$ per dataset on a validation split). **(A)** Time series: average gain in quantile loss (%) across the nine LTSF datasets. The depth-8 point is obtained by training at $R_{\text{train}} = 4$ and increasing recurrences at evaluation only. **(B)** Tabular: average gain in classification accuracy (percentage points) across classification datasets in the OpenML subset. Dashed line indicates the same-depth baseline.

and weight-tied recurrence. The deeper baseline matches a CoT model at $R = 1$ in forward-pass depth, but on some of the more modestly sized datasets we consider, this appears to encourage overfitting rather than useful additional computation. The looped baselines, by contrast, share weights across passes like our CoT models but reapply the transformer to the same token sequence rather than appending intermediate representations. They improve over the baseline but underperform CoT on average (1.9 vs. 1.6 average rank on time series, 2.5 vs. 1.6 on tabular). Together, these comparisons suggest that the gains come from the interaction of two components: weight tying (by passing through the same model multiple times) lets the model spend more compute without the increased parameter count that causes the deeper baseline to overfit, and the appended CoT tokens give the extra compute a place to store intermediate results, turning shared-weight recurrence from undifferentiated reapplication into a structured iterative computation.

## 4. Discussion

Our results show that latent chain-of-thought can transfer from language modeling to structured-data transformers. Across nine LTSF time-series datasets and 27 OpenML tabular datasets, giving a transformer multiple rounds of latent computation before it makes a prediction outperforms a same-depth no-CoT baseline, a depth-matched deeper baseline, and weight-tied looped transformers on the majority of datasets. The fact that all three baselines are dominated, which demonstrates that the gains are not attributable only to compute, parameter count, or weight tying, but to the mechanism of compressing query-position hidden states into feedback tokens and reprocessing them. The looped baselines are particularly informative since they share weights across passes but reapply the transformer to the same token sequence rather than appending an intermediate represen-

tation. Their underperformance suggests that the benefit of CoT is not iterative refinement of the hidden state per se, but the ability to write intermediate computation into a sequence that subsequent passes can attend to.

The shapes of the depth-scaling curves are also informative. On both tabular and time-series tasks, performance is non-monotonic. For tabular, a single recurrence slightly hurts on average, but additional recurrences recover and improve substantially. It is possible the depth-1 dip is a case where the model has enough additional capacity to overfit but not enough to learn a useful iterative update; deeper CoT improves by giving the model more passes. On time series, all tested depths improve over the baseline with no clear trend across $R \in \{1, 2, 4, 8\}$. The $R = 8$ result is obtained by training at $R_{\text{train}} = 4$ and increasing CoT depth only at evaluation, which indicates that the CoT update can generalize beyond its training horizon.

The most important limitation is scale. We train each model from scratch on each dataset to enable controlled comparisons, but in practice these architectures are often used with large-scale pretraining and then used out-of-the-box. Whether latent chain-of-thought helps pretrained foundation models is an empirical question that this paper does not currently answer. A second limitation is that the chain-of-thought depth $R$ is fixed per model rather than chosen adaptively per input, even though the non-monotonic performance curve suggests that the optimal depth varies across instances. An adaptive-CoT mechanism similar to PonderNet (Banino et al., 2021) would be a useful next step. Additionally, it is unclear how CoT interacts with distribution shift, which constitutes an important future direction. Finally, it would be insightful to mechanistically characterize what the feedback tokens actually encode—whether they represent residual error, prediction uncertainty, or something else.

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

## A. Architecture and training details

All models are trained from scratch on each dataset using the same optimizer and schedule. We use AdamW with learning rate $3 \times 10^{-4}$, cosine annealing, weight decay $10^{-4}$, batch size 128, and a maximum of 100 epochs with early stopping on the validation split. Hyperparameters are held constant across all models and all datasets. We do not tune per-dataset or per-model.

**Transformer backbone.** The base ($L$-layer) transformer uses $L = 4$ layers, hidden dimension 256, 8 attention heads, FFN dimension $4 \times 256 = 1024$, and dropout 0.1. The deeper baseline uses the same configuration with $2L = 8$ layers. Looped baselines reuse the same per-block configuration with $K \in \{1, 4\}$ and $M \in \{2, 4\}$.

**CoT feedback MLP.** The MLP $\phi_\theta$ that maps query-position hidden states to feedback tokens is a two-layer MLP with hidden dimension 256 and GELU activation, mapping from the transformer hidden dimension to itself.

**Tabular setup.** We use a TabICL-style three-stage architecture (Qu et al., 2025) with one column-attention layer, one row-attention layer, and a 4-layer ICL transformer at the third stage. Recurrence is applied only at the ICL stage; the column and row stages are run once per query and their outputs cached across recurrences. Predictions are trained with cross-entropy for classification and root mean squared error for regression.

**Time-series setup.** We use a patch-based forecasting transformer (Nie et al., 2023) with patch size 16, context length 1024, and the prediction horizon set per dataset (96 for ETTh1/ETTh2/ETTm1/ETTm2/ECL/Traffic/Weather/Exchange, 24 for ILI), following the standard LTSF-9 protocol (Zeng et al., 2022). Forecasts are decoded into quantile predictions at levels $\{0.1, 0.3, 0.5, 0.7, 0.9\}$ and trained with the quantile loss.

**Validation protocol.** For each dataset we hold out a validation split from the training data. The same split is used both for early stopping and for selecting the per-dataset CoT/looped depth in the "Best CoT" and "Best Looped" columns of Table 1.

## B. Datasets

**Time series (LTSF-9).** We use the nine standard long-term forecasting datasets: ETTh1, ETTh2, ETTm1, ETTm2, ECL, Traffic, Weather, Exchange, and ILI, with horizons and splits following (Zeng et al., 2022). Forecast horizon is 96 for all datasets except ILI, which uses 24.

**Tabular (OpenML subset).** We use a 27-dataset subset of OpenML comprising 15 binary classification, 8 multiclass classification, and 4 regression datasets. The full list is given in Table 2.

## C. Per-dataset results

Tables 2 and 3 report per-dataset performance for the same-depth baseline, deeper baseline, best CoT model (selected on validation across $R_{\text{train}} \in \{1, 2, 4\}$ and $R_{\text{eval}} \in \{0, 1, 2, 4, 8\}$), and best looped baseline (selected on validation across the four $K \times M$ configurations).

## D. Train-time vs. test-time recurrence

Tables 4 and 5 report performance averaged across datasets for each combination of $R_{\text{train}} \in \{0, 1, 2, 4\}$ and $R_{\text{eval}} \in \{0, 1, 2, 4, 8\}$. The $R_{\text{train}} = 0$ row is the same-depth baseline. The off-diagonal entries with $R_{\text{eval}} > R_{\text{train}}$ test whether the learned recurrent update generalizes beyond the training horizon: notably, models trained at $R_{\text{train}} = 4$ continue to perform competitively at $R_{\text{eval}} = 8$, consistent with the depth-8 result reported in Figure 2.

## E. Looped baseline breakdown

Table 1 reports the best of four looped baselines per dataset. Table 6 reports each configuration separately. The Universal-Transformer-style configurations ($K = 4$) tend to outperform the single-block-loop configurations ($K = 1$) on tabular tasks but not on time series, and no single looped configuration is consistently best.

*Table 2.* Per-dataset results on 27 datasets from OpenML (Bischl et al., 2021). Metric is AUC for binary classification, accuracy for multiclass classification, and −RMSE for regression (higher is better in all cases). Percent gain is relative to the same-depth baseline. CoT wins against the deeper baseline on 24/27 datasets.

| Dataset | Task | Baseline | Deeper | Best CoT | Best Looped | CoT % gain | Looped % gain |
|---|---|---|---|---|---|---|---|
| balance_scale | multiclass | 0.937 | 0.968 | 0.921 | 0.984 | -1.7% | +5.1% |
| breast_w | binclass | 0.978 | 0.982 | 0.981 | 0.982 | +0.3% | +0.4% |
| cmc | multiclass | 0.754 | 0.768 | 0.782 | 0.855 | +3.7% | +13.4% |
| credit_g | binclass | 0.719 | 0.700 | 0.971 | 0.781 | +35.0% | +8.5% |
| diabetes | binclass | 0.881 | 0.878 | 0.906 | 0.870 | +2.9% | -1.3% |
| tic_tac_toe | binclass | 0.682 | 0.807 | 0.922 | 0.829 | +35.2% | +21.5% |
| vehicle | multiclass | 0.894 | 0.835 | 0.850 | 0.806 | -5.0% | -10.0% |
| eucalyptus | multiclass | 0.723 | 0.764 | 0.844 | 0.763 | +16.7% | +5.6% |
| wdbc | binclass | 0.976 | 0.972 | 0.989 | 0.989 | +1.4% | +1.4% |
| banknote | binclass | 0.817 | 0.782 | 0.846 | 0.892 | +3.6% | +9.2% |
| blood_transfusion | binclass | 0.760 | 0.741 | 0.759 | 0.773 | -0.1% | +1.7% |
| kr_vs_kp | binclass | 0.949 | 0.992 | 0.978 | 0.987 | +3.0% | +4.0% |
| phoneme | binclass | 0.878 | 0.856 | 0.878 | 0.882 | +0.0% | +0.4% |
| qsar_biodeg | binclass | 0.892 | 0.900 | 0.923 | 0.879 | +3.5% | -1.4% |
| wall_robot | multiclass | 0.826 | 0.751 | 0.775 | 0.773 | -6.2% | -6.4% |
| phishing | binclass | 0.873 | 0.847 | 0.907 | 0.892 | +3.9% | +2.2% |
| steel_plates | multiclass | 0.687 | 0.651 | 0.682 | 0.641 | -0.8% | -6.7% |
| segment | multiclass | 0.762 | 0.775 | 0.805 | 0.727 | +5.7% | -4.5% |
| churn | binclass | 0.758 | 0.697 | 0.846 | 0.672 | +11.7% | -11.3% |
| electricity | binclass | 0.778 | 0.781 | 0.786 | 0.778 | +1.1% | -0.0% |
| adult | binclass | 0.817 | 0.827 | 0.831 | 0.808 | +1.7% | -1.1% |
| bank_marketing | binclass | 0.843 | 0.864 | 0.867 | 0.871 | +2.8% | +3.3% |
| jungle_chess | multiclass | 0.713 | 0.743 | 0.756 | 0.735 | +5.9% | +3.0% |
| california_housing | regression | -0.202 | -0.199 | -0.197 | -0.197 | +2.6% | +2.3% |
| abalone | regression | -2.476 | -2.416 | -2.290 | -2.251 | +7.5% | +9.1% |
| cpu_act | regression | -152.69 | -162.09 | -151.60 | -152.55 | +0.7% | +0.1% |
| superconductivity | regression | -13.35 | -12.30 | -12.25 | -14.61 | +8.3% | -9.4% |
| **Average** | | | | | | **+5.31%** | **+1.44%** |

*Table 3.* Per-dataset results on LTSF-9 time-series forecasting. Metric is mean squared error of the median forecast (lower is better). Percent gain is relative to the same-depth baseline. CoT wins against the deeper baseline on 9/9 datasets.

| Dataset | Horizon | Baseline | Deeper | Best CoT | Best Looped | CoT % gain | Looped % gain |
|---|---|---|---|---|---|---|---|
| ETTh1 | 96 | 0.4405 | 0.6687 | 0.3729 | 0.4307 | +15.3% | +2.2% |
| ETTh2 | 96 | 0.4493 | 0.5746 | 0.3386 | 0.5025 | +24.6% | -11.8% |
| ETTm1 | 96 | 0.3481 | 0.3677 | 0.3605 | 0.3134 | -3.6% | +10.0% |
| ETTm2 | 96 | 0.3273 | 0.2789 | 0.2527 | 0.2362 | +22.8% | +27.9% |
| ECL | 96 | 0.2094 | 0.2222 | 0.2012 | 0.1887 | +3.9% | +9.9% |
| Traffic | 96 | 0.4203 | 0.4196 | 0.4148 | 0.4136 | +1.3% | +1.6% |
| Weather | 96 | 0.1894 | 0.2126 | 0.188 | 0.1868 | +0.7% | +1.4% |
| Exchange | 96 | 0.208 | 0.177 | 0.1571 | 0.2434 | +24.5% | -17.0% |
| ILI | 24 | 1.344 | 1.369 | 1.217 | 1.365 | +9.5% | -1.6% |
| **Average** | | | | | | **+10.99%** | **+3.01%** |

*Table 4.* Time series: average percent gain in MSE relative to the same-depth baseline ($R_{\text{train}} = R_{\text{eval}} = 0$), across the 9 LTSF datasets, as a function of training and evaluation recurrence depth.

|                          | $R_{\text{eval}} = 0$ | 1      | 2      | 4      | 8      |
| ------------------------ | --------------------- | ------ | ------ | ------ | ------ |
| $R_{\text{train}} = 1$   | —                     | +6.2%  | —      | —      | —      |
| $R_{\text{train}} = 2$   | —                     | —      | +6.5%  | —      | —      |
| $R_{\text{train}} = 4$   | —                     | —      | —      | +5.7%  | +6.0%  |

*Table 5.* Tabular classification: average accuracy gain (percentage points) relative to the same-depth baseline, across classification datasets in the OpenML subset, as a function of training and evaluation recurrence depth.

|                          | $R_{\text{eval}} = 0$ | 1     | 2     | 4     | 8   |
| ------------------------ | --------------------- | ----- | ----- | ----- | --- |
| $R_{\text{train}} = 1$   | —                     | −1.8  | —     | —     | —   |
| $R_{\text{train}} = 2$   | —                     | —     | +3.1  | —     | —   |
| $R_{\text{train}} = 4$   | —                     | —     | —     | +6.7  | —   |

*Table 6.* Looped baseline configurations: average percent gain relative to the same-depth baseline. $K$ is the stack depth per pass; $M$ is the number of passes. "Best Looped" selects the best of the four configurations per dataset on the validation split.

|                     | $K{=}1, M{=}2$ | $K{=}1, M{=}4$ | $K{=}4, M{=}2$ | $K{=}4, M{=}4$ | Best Looped |
| ------------------- | -------------- | -------------- | -------------- | -------------- | ----------- |
| Time series (n=9)   | -0.5%          | -8.6%          | +0.5%          | -10.1%         | +3.0%       |
| Tabular (n=27)      | -2.2%          | -1.7%          | -0.4%          | -1.7%          | +1.4%       |

