# OpenReview forum: "Latent Chain-of-Thought Improves Structured-Data Transformers"
_ICML.cc/2026/Workshop/FMSD — FMSD @ ICML 2026 Poster_

### Official Review · Reviewer_54Wa · 2026-05-13

**Rating:** 3
**Confidence:** 4

**Review:**

**Summary:**
The authors propose an adaptation of a technique that autoregressively compresses and appends latent feedback tokens prior to final prediction, and evaluate its impact on model performance for classification and forecasting tasks.

**Strengths:**
- Novel adaptation of latent feedback recurrence techniques from language modeling to structured data (time-series forecasting and tabular classification).
- The proposed framework generalizes across transformer architectures.
- The technique is evaluated across 9 time-series forecasting datasets and 27 tabular classification datasets.
- Current results demonstrate promising empirical gains over baselines across both tasks.


**Areas for Improvement:**

1.) Chain-of-thought concept/definition:

- Chain-of-thought is a generalization capability, not a mechanism for generating intermediate tokens or scaling test-time compute. The paper seems to conflates the phenomenon with the mechanism used to induce it. I recommend clarifying the definition of and discussion around chain-of-thought reasoning.

- It seems the technique itself is more precisely described as latent feedback recurrence than chain-of-thought reasoning. Clarifying the specific technique and terminology would help position the claims and justify the focus of the related work, given the extensive literature on chain-of-thought reasoning in LLMs [1].

- For example, the following claim should be revisited: "Our results show that latent chain-of-thought can transfer from language modeling to structured-data transformers." (Lines 204-205). It is not chain-of-thought that is being transferred, but rather a specific technique, and the empirical takeaway is performance gains rather than evidence of chain-of-thought reasoning.

    [1] Reasoning Beyond Language: A Comprehensive Survey on Latent Chain-of-Thought Reasoning

2.) Formal Problem Setup:
- The methods should be more clearly organized and outlined with formal notation. For example, the prediction task lacks formal introduction (i.e., inputs, model, outputs, and loss are not specified with notation). Separately outlining the forecasting and classification tasks (datasets, evaluation metrics, etc.) would be helpful.

- Defining structured data would be helpful for readers. Such term is not typically used in forecasting literature.

3.) Clarity of Contributions
- The paper could benefit from a clearer delineation of the novel contributions relative to prior work. The technique appears to adapt existing LLM methods to forecasting and classification (also for Transformer architectures), but it's unclear how the technique specifically differs from prior work.


**Justification of Score:**

The work presents an interesting empirical contribution but would benefit from another round of revision before presentation, particularly regarding the three areas identified above to clarify the scope, claims, and contributions of the work.

---

### Official Review · Reviewer_AUgz · 2026-05-16

**Rating:** 7
**Confidence:** 4

**Review:**

Summary of contributions

This paper investigates whether latent chain-of-thought (CoT) can improve transformers for structured data, including time-series forecasting and tabular prediction. The authors propose a recurrent latent reasoning framework in which query-position hidden states are compressed into feedback tokens and appended back to the input sequence for multiple rounds of recurrent processing. Experimental results show that latent CoT consistently improves performance across both domains and outperforms alternative recurrence-based baselines.

Strengths
1. Interesting and timely research direction. The paper explores whether chain-of-thought style iterative computation, which has recently shown strong success in language models, can transfer to structured-data transformers. Extending latent reasoning concepts beyond language modeling is an important and novel research question.
2. Strong experimental design and baselines. The paper carefully compares against same-depth, deeper, and looped-transformer baselines, allowing the authors to identify the contribution of latent CoT tokens. This significantly strengthens the empirical claims.
3. Consistent empirical improvements. The method demonstrates gains across the majority of datasets in both forecasting and tabular prediction settings, and the results suggest that latent CoT provides benefits beyond simply scaling model depth or parameter count.

Weaknesses
1. The mechanism behind the improvements remains insufficiently understood. Although the paper demonstrates empirical gains, it remains unclear what information the latent feedback tokens actually encode and whether they correspond to meaningful iterative reasoning rather than generic hidden-state refinement.
2. The computational overhead may limit practical applicability. Each recurrence requires rerunning the transformer over an increasingly longer sequence due to appended CoT tokens, leading to additional inference latency and memory cost. The paper focuses mainly on prediction quality while providing limited analysis of the tradeoff between performance gains and increased test-time compute.
3. The evaluation is limited to relatively small-scale supervised settings. All models are trained from scratch on individual datasets rather than evaluated in large-scale pretrained TSFM or tabular foundation model settings, making it unclear whether the proposed latent CoT mechanism would remain effective at foundation-model scale.

Suggestions
1. The paper would be strengthened by additional mechanistic analysis of the latent feedback tokens, such as probing whether they encode uncertainty or intermediate task-specific representations. In particular, visualizing and analyzing the attention maps between the original query tokens and the appended CoT tokens across recurrence steps could provide more insight into how the model utilizes the latent reasoning tokens during iterative computation.
2. More detailed efficiency analysis, including FLOPs, inference latency, and memory consumption across recurrence depths, would help clarify the practical cost-benefit tradeoff of latent CoT.
3. Evaluating the proposed framework in large-scale pretrained structured-data foundation models would significantly improve the practical relevance and impact of the work.
4. It would be interesting to explore adaptive recurrence strategies that dynamically determine the required amount of latent reasoning per sample, rather than using a fixed CoT depth for all inputs.

---

### Official Review · Reviewer_viWx · 2026-05-18

**Rating:** 7
**Confidence:** 3

**Review:**

## Summary
The paper investigates whether latent chain-of-thought style computation can improve transformers for structured data tasks. The proposed approach introduces recurrent latent computation by compressing hidden states at prediction positions into feedback tokens, appending them to the input, and processing the sequence again through the transformer. The paper compares this approach against same-depth baselines, deeper compute-matched baselines, and looped transformers without explicit feedback tokens. Experiments on time series forecasting and tabular prediction datasets show that the proposed method consistently improves performance over the baselines on average.

## Strengths
- The paper explores an interesting idea that appears to be a natural next step in the design of foundation models for structured data. - The study is principled and the results are promising.
- The experiments are well designed and directly address the main research questions. I also appreciated that the evaluation covers both time series and tabular tasks.
- I especially appreciated the thoughtful discussion of limitations in Section 4.

## Ideas for Improvement
- The LTSF datasets have been extensively criticized, and there are now many stronger alternatives for evaluation (e.g. tasks from GIFT-Eval, TIME, Monash, or fev-bench). In the current setup, the four ETT datasets also contribute disproportionately to the aggregate results. I recommend the authors consider the discussion in "Seeking SOTA: Time-Series Forecasting Must Adopt Taxonomy-Specific Evaluation to Dispel Illusory Gains" by Saqur et al., which provides a recent critique of LTSF-style evaluations.
- An interesting open question is how well the proposed approach generalizes across domains and tasks. I imagine this may be addressed in a future conference version of the work, but I wanted to flag it as an important direction for further study.

## Justification
The paper proposes an interesting and relevant idea, supported by a solid experimental study and thoughtful analysis. Overall, I believe the work is a good fit for the workshop and recommend acceptance.